

# Decoupling peroxyacetyl nitrate from ozone in Chinese outflows observed at Gosan Climate Observatory

7          Jihyun Han[1*], Meehye Lee[1], Gangwoong Lee[2], Louisa K. Emmons[3]

[1]Department of Earth and Environmental Sciences, Korea University, Seoul,

Republic of Korea

[2]Department of Environmental Science, Hankuk University of Foreign Studies, Yongin,

Republic of Korea

[3]Atmospheric Chemistry Observations and Modeling Laboratory, National Center for

Atmospheric Research (NCAR), Boulder, CO, USA

*now at: Korea Environment Institute, Sejong, Republic of Korea

19      Correspondence to: M. Lee (meehye@korea.ac.kr )

23              Submitted to Atmospheric Chemistry and Physics

24                          December 2016





**Abstract**
We measured peroxyacetyl nitrate (PAN) and other reactive species such as $O_3$, $NO_2$, CO,
and $SO_2$ with aerosols including $PM_{10}$ and $PM_{2.5}$ organic carbon (OC) and elemental carbon
(EC) at Gosan Climate Observatory in Korea (33.17°N, E126.10°E) during October 10 to
November 6, 2010. PAN was determined through fast gas chromatography with luminol
chemiluminescence detection at 425 nm every 2 min. The PAN mixing ratios ranged from 0.1
(detection limit) to 2.4 ppbv with a mean of 0.6 ppbv. For all measurements, PAN was
unusually better correlated with $PM_{10}$ (Pearson correlation coefficient, $\gamma = 0.75$) than with $O_3$
($\gamma = 0.67$). In particular, the $O_3$ level was highly elevated with $SO_2$ at midnight, along with a
typical midday peak when air was transported rapidly from the Beijing areas. The PAN
enhancement was most noticeable during the occurrence of haze under stagnant conditions.
In Chinese outflows slowly transported over the Yellow Sea, PAN gradually increased up to
2.4 ppbv at night, in excellent correlation with a concentration increase of $PM_{2.5}$ OC and EC,
$PM_{1.0}$ $K^+$, and $PM_{10}$ mass. The high $K^+$ and OC/EC ratio indicated that the air mass was
impacted by biomass combustion. This study highlights PAN decoupling with $O_3$ in Chinese
outflows and suggests PAN as a potential indicator of overall aerosol formation in aged air
masses impacted by biomass burning.

Key words: PAN, $O_3$, $PM_{10}$, Chinese outflow, Haze, Biomass combustion





## 1. Introduction

At the surface, ozone is primarily photochemically produced, and the contribution from the stratosphere is generally small. Ozone is formed through reactions of various precursors such as CO, $CH_4$, volatile organic compounds (VOCs), and $NO_x$ (e.g., Brasseur et al., 1999; Jacob, 2000; Nielsen et al., 1981). Likewise, peroxyacetyl nitrate (PAN) is a secondary product of urban air pollution and a significant oxidant in the atmosphere (e.g., Hansel and Wisthaler, 2000; La Franchi et al., 2009; Lee et al., 2012; Liu et al., 2010; Roberts et al., 2007). PAN is solely produced by the photochemical reaction between the peroxyacetyl radical and nitrogen dioxide, and the peroxyacetyl radical is derived from the OH oxidation or photolysis of VOCs such as acetaldehyde, methylglyoxal, and acetone (e.g., Fischer et al., 2014; La Franchi et al., 2009; Lee et al., 2012). For this reason, PAN is a very useful indicator of photochemical air pollution. As thermal decomposition is a major PAN sink in the troposphere (Beine et al., 1997; Jacob, 2000; Kenley and Hendry, 1982; Talukdar et al., 1995), the lifetime of PAN depends on temperature. For example, the PAN lifetime is ~5 years at −26°C and 1 h at 20°C (Fischer et al., 2010; Zhang et al., 2011). At high altitudes above ~7 km, photolysis becomes the most important loss process for PAN (Talukdar, et al., 1995). Thus, PAN can be an indicator of $NO_y$ concentration in the free troposphere in urban areas and a guide for the long-range transport of $NO_x$ in remote regions (Jacob, 1999).

In the past decades, PAN was measured not only in urban areas (Aneja et al., 1999; Gaffney et al., 1999; Grosjean et al., 2002; Lee et al., 2008; Zhang et al., 2014) but also in background regions (Fischer et al., 2011; Kanaya et al., 2007; Lee et al., 2012), onboard aircraft (Tereszchuk et al., 2013), and ships (Roberts et al., 2007). PAN concentrations were in the range of a few ppbv in urban areas close to VOCs and $NO_x$ sources (Lee et al., 2008; Zhang et al., 2011). In remote regions, PAN mixing ratios were generally in the range of a




few pptv (Gallagher et al., 1990; Mills et al., 2007; Muller and Rudolph, 1992; Staudt et al.,

70    2003).

In recent years, $NO_x$ and VOCs have gradually increased in East Asia, particularly China
(Akimoto, 2003; Liu et al., 2010; Ohara et al., 2007), leading to an increase in the
concentrations of photochemical byproducts such as PAN and $O_3$ not only in East Asia (Liu et
al., 2010; Wang et al., 2010; Zhang et al., 2009; Zhang et al., 2011; Zhang et al., 2014) but
also in North America (Fischer et al., 2010; Fischer et al., 2011; Jaffe et al., 2007; Zhang et
al., 2008). These results were also demonstrated by the GEOS-Chem model (Zhang et al.,
2008). In addition to urban plumes, PAN was reported to be enhanced by biomass
combustion (Alvarado et al., 2010; Coheur et al., 2007), such as open burning and use of
biofuel, which is used to take place often in China after crop harvesting (Cao et al., 2006;
Duan et al., 2004). In this context, PAN is a useful indicator for diagnosing Chinese outflows
and assessing their perturbation on regional air quality in the northwestern Pacific region.
Gosan Climate Observatory (GCO) is an ideal place to monitor Asian outflows and their
transformation and to estimate their impact on air quality over the northern Pacific region
(Lee et al., 2007; Lim et al., 2012). In the present study, PAN was first measured
continuously at GCO to characterize its variation and source in relation to $O_3$ and to
understand the influence of Chinese outflows on the regional air quality.

**2. Experiments**
PAN measurements were conducted at GCO (33.17°N, E126.10°E) on Jeju Island from
October 10 to November 6, 2010. GCO is located on a cliff at the western edge of Jeju Island.
PAN was determined through fast gas chromatography (GC) with luminol
chemiluminescence detection, which is described in detail elsewhere (Gaffney et al., 1998;





Lee et al., 2008; Marley et al., 2004). Here, we briefly describe the measurement method.
Ambient air PAN and $NO_2$ (and peroxypropyl nitrate (PPN) if present) were separated along
a 10-m capillary GC column (DB-1, J&W Scientific, Folsom, CA, USA), whose end was
connected to a luminol cell where the column effluent reacted with luminol, giving off
luminescent light (Lee et al., 2008; Lee et al., 2012). The concentrations of PAN and other
species were determined from the chemiluminescence signals detected by a gated photon
counter (HC135-01, Hamamatsu, Bridgewater, NJ, USA) at 425 nm, which was set at 800 V
and operated at room temperature (Gaffney et al., 1998; Lee et al., 2012; Lee et al., 2008).
PAN was calibrated against standards synthesized by the nitration of peracetic acid in n-
tridecane (Gaffney et al., 1984; Gregory, 1990). The nominal detection limit of PAN defined
by $3\sigma$ of the lowest standard was 100 pptv (Lee et al., 2008).
Water-soluble ions of $PM_{1.0}$ were collected by a particle-into-liquid sampler and analyzed
by ion chromatography. Gaseous species, including $O_3$, $NO$, $NO_2$, $CO$, and $SO_2$, were
measured by UV absorption, chemiluminescence with a molybdenum converter, non-
dispersive infrared, and pulse UV fluorescence method, respectively (NIER, 2016). Aerosol
species, including $PM_{10}$ mass and $PM_{2.5}$ OC and EC were measured and recorded along with
meteorological parameters (relative humidity, temperature, and wind speed). The detailed
results of the aerosol measurements can be found in Shang et al. (2017).
The three-day backward trajectories of air parcel at 850m a.s.l. for every one hour were
calculated using NOAA Air Resources Laboratory (ARL) Hybrid Single-Particle Lagrangian
Integrated Trajectory (HYSPLIT) model (version 4) (Draxler and Rolph, 2012; Rolph, 2012,
http://www.arl.noaa.gov/ready/hysplit4.html).

**3. Results**





In the present experiments, PAN mixing ratios range from 0.1 to 2.4 ppbv, with an average
of 0.6 ppbv. This mean value is lower than those observed in other Asian megacities: Beijing
(1.41 ppb in the summer), Pearl River Delta region (1.32 ppb in the summer), and Seoul (0.8
ppb in the early summer); similar to those of suburban areas in China, e.g., Lanzhou (0.76
ppb in the summer); and higher than those in the western coast of the US, e.g., Sacramento
(0.45 ppb in the summer), Mt. Bachelor (0.144 ppb in the spring and early summer), off the
western coast of the US (0.65 ppb in the spring), and over the remote North Pacific (total
PAN < 0.3 ppb in spring) (Bertram et al., 2013; Fischer et al., 2011; La Franchi et al., 2009;
Lee et al., 2008; Roberts et al., 2004; Wang et al., 2010; Zhang et al., 2009; Zhang et al.,
2011). Because the PAN lifetime is greatly dependent on temperature, its concentration
decreases with increasing distance from the source regions. The PAN mixing ratios calculated
in this study thus lie in-between the levels for the East Asian megacities and the northern
Pacific. The distributions of all measured species, including PAN and $O_3$, are presented in Fig.
1. In particular, there are several periods characterized by high concentrations of PAN, $O_3$,
and $PM_{10}$. In terms of PAN, four periods are particularly interesting (Fig. 1). High $O_3$
concentrations were observed during October 31–November 2 [episode 1] but did not
coincide with high PAN concentrations. During October 28–29 [episode 2], $NO_2$ was
noticeably increased. On the other hand, PAN and $O_3$ concentrations were both high during
October 20–21 [episode 3] and November 4–5 [episode 4]. Episodes 3 and 4 are
characterized by haze, while episodes 1 and 2 are characterized by urban influence in the
Korean and Beijing outflows, respectively.
In the present study, PAN correlates reasonably well with $O_3$ ($\gamma = 0.67$) and even better
with $PM_{10}$ ($\gamma = 0.75$). In general, $O_3$ and PAN exhibit typical diurnal variation with a
maximum recorded in the afternoon, which results in a good correlation between the two



(Brasseur et al., 1999; Gaffney et al., 1999; Ridley et al., 1990; Schrimpf et al., 1995; Wang
et al., 2010). In this study, however, the $O_3$ peak was often found in the early morning and
late afternoon for several days (Fig. 1). Observing the diurnal variations in the entire PAN
concentration measurement set (Fig. 2), the maximum was clearly recorded in the morning
with the highest outliers, which is rather similar to that of $PM_{10}$. The diurnal pattern of $NO_2$
shows little variation, even though its concentrations were increased in the morning along
with PAN. This first measurement of PAN at GCO reveals that PAN is not always coupled
with $O_3$, which was not typically observed at remote sites in previous studies (e.g., Fischer et
al., 2010; Lee et al., 2012).

**4. Discussion**
**4.1. Decoupling of PAN from $O_3$**
To examine the detailed mechanism of the decoupling of PAN from $O_3$, the daily
maximum concentrations of PAN and $O_3$ were further explored. The recorded daily PAN
maxima were generally in good correlation with $O_3$, albeit the relationship did not seem to
hold at high concentrations of PAN and $O_3$ (Fig. 3). The daily maxima were then categorized
into four groups according to the time when each $O_3$ and PAN maximum was recorded: "$O_3$
day-PAN day," "$O_3$ day-PAN night," "$O_3$ night-PAN day," and "$O_3$ night-PAN night." The
day interval started from 08:00 and ended at 18:00 (local time), based on the times of sunrise
and sunset during the experiment period. While the high PAN concentrations were associated
with the "$O_3$ day-PAN day" group (cross symbols in Fig. 3), the enhanced $O_3$ concentration
was recorded in the "$O_3$ night-PAN night" group (star symbols in Fig. 3). The "$O_3$ night-PAN
night" group unexpectedly held more data points than the "$O_3$ day-PAN day" group, even
though the "$O_3$ night-PAN night" group concentrations were lower (Fig. 3). In addition, there



were several days classified in the "$O_3$ night-PAN day" (marked by diamond) and "$O_3$ day-
PAN night" groups, but with less frequency and lower concentrations. These results indicate
that the decoupling of PAN from $O_3$ was primarily due to the elevated concentrations of $O_3$
and/or PAN at night. The four high PAN and $O_3$ episodes identified in this study fall under
the category of "$O_3$ night-PAN night" or "$O_3$ day-PAN day." This point will be further
examined to identify the chemical and physical processes responsible for PAN being
decoupled from $O_3$, instead of being coupled with $PM_{10}$. The overall characteristics of the
four episodes are summarized in Table 1.

**4.2. Export of $O_3$ from Asian continents (episodes 1 & 2)**
High $O_3$ concentrations were encountered around midnight on three consecutive days from
October 31 to November 2 (episode 1), during which $SO_2$ reached its maximum
concentration (Fig. 1). The backward trajectories of air masses revealed that air passed
through the Beijing area during this period (Fig. 4). The wind was strong (13.5 m/s on
average) and the recorded $O_3$ maximum (80.6 ppbv) was concurrent with the PAN maximum
(0.9 ppbv) around midnight on November 1st (Fig. 4).
All these results indicate that the air was heavily influenced by outflow from the Beijing
area, as previously hypothesized (Lim et al., 2012), and that the nighttime enhancement of $O_3$
and PAN resulted from the fast transport of relatively less-aged urban plumes. In this episode,
PAN and $O_3$ could have been formed in urban areas or produced in the outflow while being
transported. Because the overall correlation between $O_3$ and PAN was the best with the
highest daily $\Delta O_3/\Delta PAN$ among all cases discussed in this study, episode 1 likely represents
an event of rapid transport from the Beijing area (Fig. 5).
In previous studies, the nighttime enhancement of $O_3$ was observed at GCO (e.g., Lee et
al., 2007) in association with pollutant-laden air coming from Beijing. Similarly, Banta et al.





(1998) pointed out that the evening $O_3$ maximum was due to long-range transport of $O_3$ from
nearby urban areas. Wang et al. (2011) reported that the $O_3$ lifetime was about two days in
East China during the summer, which is sufficient time for air to travel to GCO. Therefore,
the nighttime maximum of $O_3$ can be attributed to the export of $O_3$ from megacities in China,
causing PAN to be decoupled from $O_3$. Another night maximum of $O_3$ was recorded on
October 29. Note that $NO_x$ was highly elevated during October 28–29 (episode 2) (Fig. 5b).
In contrast, $O_3$ and PAN levels remained relatively low, leading to the lowest daily
$\Delta O_3/\Delta PAN$ among all episodes. In this case, air masses passed through the Korean Peninsula,
carrying low $O_3$ being titrated by high $NO_x$ (Brasseur et al., 1999; Jacobson, 2005).

**4.3.  PAN enhancement upon occurrence of haze (episodes 3 & 4)**
In this study, two haze events were observed in the very beginning (October 20–21;
episode 3) and the end of the study period (November 4–5; episode 4). The first haze event
occurred on October 18th and lingered until October 21st, during which $O_3$ concentrations
were gradually elevated. A second peak was recorded around midnight of October 19th and
20th, and the maximum was reached in the afternoon of October 20th (Figs. 1 and 3). In this
episode, the maximum concentrations of $O_3$ and PAN were 78.9 ppbv and 2.0 ppbv,
respectively, on October 29th, when the highest $NO_2$ concentration (12.7 ppbv) was observed
under low wind speed (6.6 m/s daily average). The air mass trajectories suggest the influence
of the Korean Peninsula, particularly the Seoul metropolitan area, in addition to East China
(Fig. 4b).
In the second haze event (episode 4), an air mass was slowly transported from East China,
including the Jiangsu province, under stagnant condition which was developed by an
anticyclone system (Fig. 4). We measured the highest concentrations of all aerosol species



including the $PM_{10}$ mass as well as PAN and $O_3$, which were 170 µg/m$^3$, 2.4 ppbv, and 87.5
ppbv, respectively. Other reactive gases such as CO, $SO_2$, and $NO_2$ were also highly elevated.
Note that PAN and $O_3$ gradually increased through the night, leading to a nighttime maximum
of both species on November 4th. It is likely that the pre-formed PAN and $O_3$ were
continuously transported into Gosan at night.
In section 4.2, the nighttime $O_3$ peak was attributed to the transport from nearby urban
areas to Jeju Island. The two haze episodes were also observed in continental outflows.
Unlike $O_3$, however, PAN is linearly correlated with the $PM_{10}$ mass and major constituents of
aerosols including $PM_{2.5}$ OC, $PM_{2.5}$ EC, and $PM_{1.0}$ K$^+$, whose concentrations were
remarkably high in this episode.
PAN is formed through the reaction of the peroxyacetyl radical and nitrogen dioxide (Eq. 1)
and decomposed at high temperature (Eq. 2), returning these radicals. Unless the NO
concentration is high (Eq. 3), the peroxyacetyl radical recombines with $NO_2$, producing PAN.
Thus, the total lifetime of PAN depends on the $NO_2$/NO ratio and temperature (Eq. 4)
(Brasseur et al., 1999).
$$CH_3C(O)O_2 + NO_2 + M \rightarrow PAN + M \quad (1)$$
$$PAN \rightarrow CH_3C(O)O_2 + NO_2 \quad (2)$$
$$CH_3C(O)O_2 + NO \rightarrow CH_3CO_2 + NO_2 \quad (3)$$
$$T_{eff} = T_d \left(1 + \frac{k_1[NO_2]}{k_2[NO]}\right) \ [\text{sec}^{-1}] \quad (4)$$
where $T_d$ and $T_{eff}$ indicate the lifetime against decomposition and the effective lifetime of
PAN (Brasseur et al., 1999). The effective lifetime of PAN was estimated through Eq. 4 using
the rate constants proposed by Brasseur et al. (1999), Jacobson (2005), and Maricq and
Szente (1996).
During the haze event, NO was close to the detection limit, while $NO_2$ was greatly
enhanced. Owing to the high $NO_2$/NO ratio, the effective lifetime of PAN increased by 57





times; this possibly contributed to the gradual increase in PAN through the night on
November 4th. In an aged plume, $NO_2$ is likely to be recycled with $O_3$ during the day and
PAN during the night. Fischer et al. (2014) also reported that, at night, PAN can be produced
from the reaction of acetaldehyde with the nitrate radical.

Besides $PM_{10}$, PAN was also well correlated with $PM_{2.5}$ OC and EC not only during this

haze episode but also during the entire measurement period (Fig. 6b and c). Furthermore, the
enhancement of PAN was concurrent with that of OC and $K^+$, resulting in excellent
correlation between them (Fig. 6e and f). In fact, the $\Delta OC/\Delta EC$ ratio of episode 4 was much
higher (7) than those of the other episodes (~2.5) (Fig. 6d). The fraction of $PM_{2.5}$ against
$PM_{10}$ was also the highest in this episode, indicating significant contribution of secondary
aerosols. These observations suggest that air masses were affected by biomass combustion
(e.g., Ram et al., 2008, 2012; Saarikoski et al., 2008).

According to previous studies, PAN can be produced in plumes through biomass

combustion (Alvarado et al., 2010; Coheur et al., 2007; Liu et al., 2016; Tereszchuk et al.,
2013). In northeast China, open burnings related to agricultural activities frequently occur
during the spring and fall (Duan et al., 2004; Yang et al., 2005). Kudo et al. (2014) also
reported that, upon burning crop residue in Yangtze region, the levels of oxygenated VOCs
were elevated together with $NO_x$. In addition, biofuel is used for cooking and heating and as
an energy source in China's industry (Cao et al., 2006).

Therefore, PAN is likely to increase when haze occurs and fine aerosols are transformed as

air masses carrying combustion emissions are slowly transported from China over the Yellow
Sea. Additionally, the results of this study imply that PAN can be used as a robust tracer for
continental outflows in northeast Asia, to identify transport- and chemical transformation-
dominant regimes. In a transport-dominant regime, $O_3$ export was distinguished by the



highest levels of primary gaseous species such as $SO_2$ and relatively low levels of PAN. In
contrast, fine aerosol species are enhanced in a chemical transformation regime, leading to
haze events with relatively more enhanced PAN compared to $O_3$.
Finally, the measured $O_3$ and PAN concentrations were compared to results from a global
chemistry model, the Community Atmosphere Model with Chemistry (CAM-Chem), a
component of the Community Earth System Model (CESM) (Lamarque et al., 2012; Tilmes
et al., 2015). The CAM-chem results shown here follow the configuration used for the
HTAP2 (Hemispheric Transport of Air Pollution, Phase 2) intercomparison (e.g., Stjern et al.,
2016). CAM-chem is nudged to observed meteorology (GEOS-5) to reproduce the actual
period of the observations (Oct 2010). The emissions used in the model are the HTAP2
inventory (Janssens-Maenhout, et al., 2015), which include the "MIX" Asian emissions
inventory. Biomass burning emissions are from the Global Fire Emissions Database (GFED3)
(Randerson et al., 2013). In the model simulation, $O_3$ and PAN were highly underestimated
during the episodes observed in Chinese outflows, although the variation around average
level of $O_3$ and PAN was well captured (Fig. 7). The enhancement of PAN during the haze
events was not well represented in the model (Oct 20–21 and Nov 4–5). The timing of the $O_3$
diurnal variability was captured by the model, although the magnitude of the variation was
underestimated. These results reveal that the current understanding of Chinese outflow is still
not sufficient, thereby causing uncertainty in estimating its effect on air quality in the
northwestern Pacific Rim.


**5.    Conclusions**
The first measurements of PAN, reactive gases, and aerosol species were conducted at
GCO during October 19 to November 6, 2010. The average concentration of PAN was 0.6
ppbv with a maximum of 2.4 ppbv, which was lower than those in major cities in East Asia
but much higher than the background concentrations in other regions. In addition, PAN and
$O_3$ concentrations were well correlated ($\gamma = 0.67$). However, the comparison of the daily
maxima of PAN and $O_3$ highlighted that they were not proportionally enhanced. That is,
either PAN was relatively more elevated than $O_3$ or the highest $O_3$ was associated with low
levels of PAN. Unexpectedly, both PAN and $O_3$ often reached their maxima at night. As a
result, PAN was decoupled from $O_3$ and better correlated with the $PM_{10}$ mass ($\gamma = 0.75$) than
with $O_3$. In this study, these high-concentration episodes were all encountered in association
with continental outflows, and thus, two high-$O_3$ and two high-PAN events were recorded
and investigated in detail.
During the $O_3$ episodes, both $O_3$ and PAN concentrations reached their maximum values at
night. In episode 1 (Oct. 31 to Nov. 2), the $O_3$ concentration was increased to 80.6 ppbv, with
a high $SO_2$ concentration under strong wind. It was a typical Beijing plume observed in the
study region. In comparison, $NO_2$ was greatly increased in episode 2 (Oct. 28–29) when the
air masses were affected by urban emissions from Korean Peninsula. Although the maximum
$O_3$ level was lower during episode 2, these two cases demonstrated well how $O_3$ was exported
from the East Asian continent.
The remaining two episodes were highlighted by enhanced PAN concentrations and
characterized by haze occurrence. During episode 3 (Oct. 20–21), PAN and $O_3$ concentrations
increased up to 2.0 ppbv and 78.9 ppbv, respectively, with high NOx levels, probably
influenced by emissions from Korea. Episode 4 (Nov. 4–5) was characterized by the highest





concentrations of almost all measured species, including PAN, $O_3$, $PM_{10}$ mass, and $PM_{1.0}$
species; the maximum recorded concentrations of PAN, $O_3$, and $PM_{10}$ mass during this
interval were 2.4 ppbv, 87.5 ppbv, and 170 $\mu g/m^3$, respectively. Note that, along with $PM_{10}$
and $O_3$, PAN was gradually increased through the night. In this episode, an air mass was
slowly transported from eastern China. With depleted NO, the effective lifetime of PAN was
greatly extended. In addition, PAN concentration showed good correlation with OC, EC, and
$K^+$; in fact, the correlation of PAN with $K^+$ was comparable to that of OC with $K^+$. These
results, in conjunction with the high $\Delta OC/\Delta EC$ (7), imply that the observed haze was mainly
caused by the emissions produced by biomass combustion. These results suggest that PAN is
a useful tool for distinguishing continental outflows that were typically observed in northeast
Asia.
The comparison between the measured and calculated concentrations using the CAM-
Chem-HTAP2 model showed that the model underestimated the $O_3$ and PAN levels in
Chinese outflows, particularly for haze incidence. These results reveal that Chinese outflows
are still poorly understood and not well captured in the model.

**Acknowledgments**
This study was funded by the Korea Meteorological Administration Research and
Development Program under Grant KMIPA 2015-6020. The National Center for Atmospheric
Research is funded by the National Science Foundation. The authors gratefully acknowledge
the NOAA Air Resources Laboratory (ARL) for the provision of the HYSPLIT transport and
dispersion model and/or READY website (http://www.ready.noaa.gov) used in this
publication.



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





**Tables**

Table 1. Chemical and meteorological characteristics of the four episodes.

|  | Episode 1 | Episode 2 | Episode 3 | Episode 4 |
|---|---|---|---|---|
| Type | Transport dominant | Transport dominant | Chemical transformation | Chemical transformation |
| Event | $O_3$ export | $O_3$ export | Haze | Haze |
| $O_3$ (ppbv) | 60.2 (80.6) | 45.6 (62.8) | 59.7 (78.9) | 61.8 (87.5) |
| PAN (ppbv) | 0.5 (0.9) | 0.5 (0.8) | 1.2 (2.0) | 1.3 (2.4) |
| $PM_{10}$ (µg/m$^3$) | 57 (90) | 38 (56) | 69 (96) | 100 (170) |
| $SO_2$ (ppbv) | 4.3 (12.9) | 2.0 (4.4) | 2.6 (5.4) | 4.4 (9.5) |
| $NO_2$ (ppbv) | 3.7 (7.3) | 6.2 (12.1) | 6.2 (12.7) | 6.1 (9.9) |
| Wind Speed (m/s) | 13.5 (16.0) | 9.5 (16.1) | 6.6 (10.2) | 5.0 (7.7) |

*Measurements are given for the average with the maximum in the parenthesis.





**Figure Captions**

Figure 1. Temporal variations of measured species, including PAN, $PM_{10}$, $O_3$, $NO_2$, NO, $SO_2$,
and, CO, and meteorological parameters, including relative humidity, temperature,
and wind speed in fall 2010. Episodes 1–4, described in the main text, are shaded
in blue and yellow.

Figure 2. Diurnal variations in the concentrations of $O_3$, $NO_2$, PAN, and $PM_{10}$, measured at
GCO in the fall of 2010 (5 min data of $O_3$, $NO_2$, 2 min data of PAN, and 1 h data of
$PM_{10}$).

Figure 3. Comparison of $O_3$ with the PAN daily maxima. The time when the daily maximum
appears is classified as daytime (08–18 h) and nighttime (the rest) based on the
time of sunrise and sunset. Numerals indicate the days.

Figure 4. The three-day NOAA HYSPLIT backward trajectories of air masses for every one
597        hour observed at GCO during episode 1 (Oct. 31to Nov. 2), episode 2 (Oct. 28–29),
episode 3 (Oct. 20–21), and episode 4 (Nov. 4–5). They are colored according to
the level of (a) PAN, (b) $O_3$, and, (c) $NO_2$ at GCO at the time of the trajectory
initialization. The trajectories north of 50°N are not shown.

Figure 5. Correlations among PAN, $PM_{10}$, $O_3$, and carbonaceous compounds in $PM_{2.5}$: (a) $O_3$
and PAN, (b) $NO_2$ and PAN, and (c) $O_3$ and PAN. The red lines in (a) and (b)
represent linear regression for episode 4.

Figure 6. Correlations among PAN, $K^+$ ion of $PM_{1.0}$, and carbon components of $PM_{2.5}$ for
three cases: (a) $PM_{10}$ and PAN, (b) $PM_{2.5}$ OC and PAN, (c) $PM_{2.5}$ EC and PAN, (d)
$PM_{2.5}$ OC and EC, (e) $PM_{1.0}$ $K^+$ and PAN, and (f) $PM_{1.0}$ $K^+$ and $PM_{2.5}$ OC. The red
lines represent linear regression for episode 4.

Figure 7. Comparison between the observed and calculated (a) PAN and (b) $O_3$
concentrations by CAM-chem model.






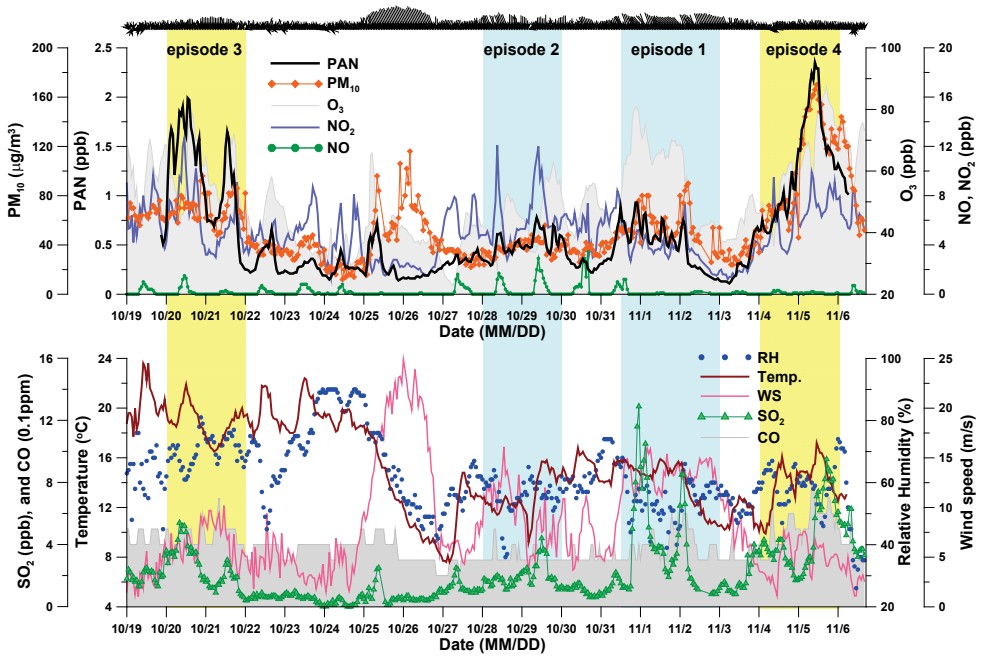



Figure 1.






Figure 2.





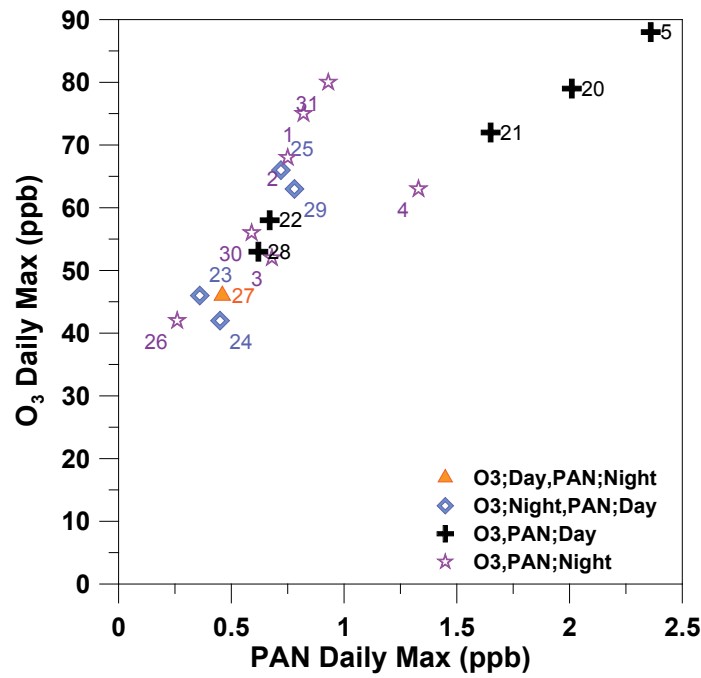


Figure 3.





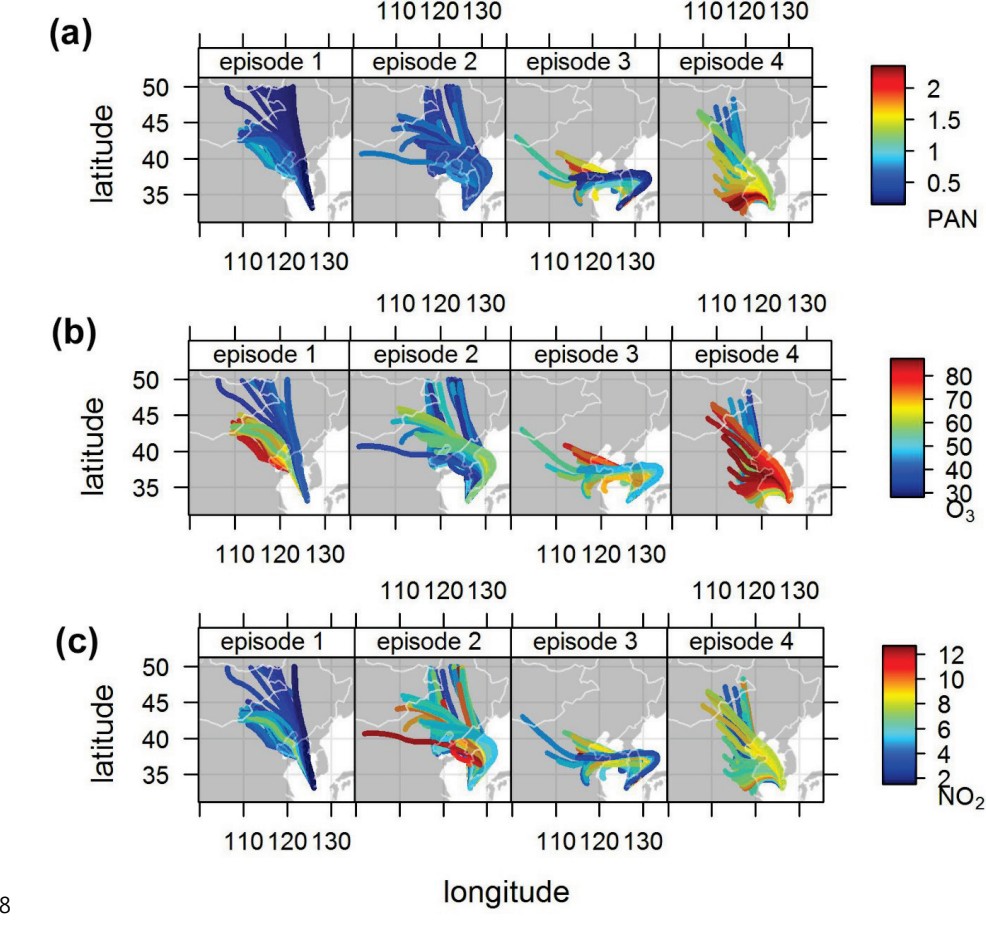


Figure 4.





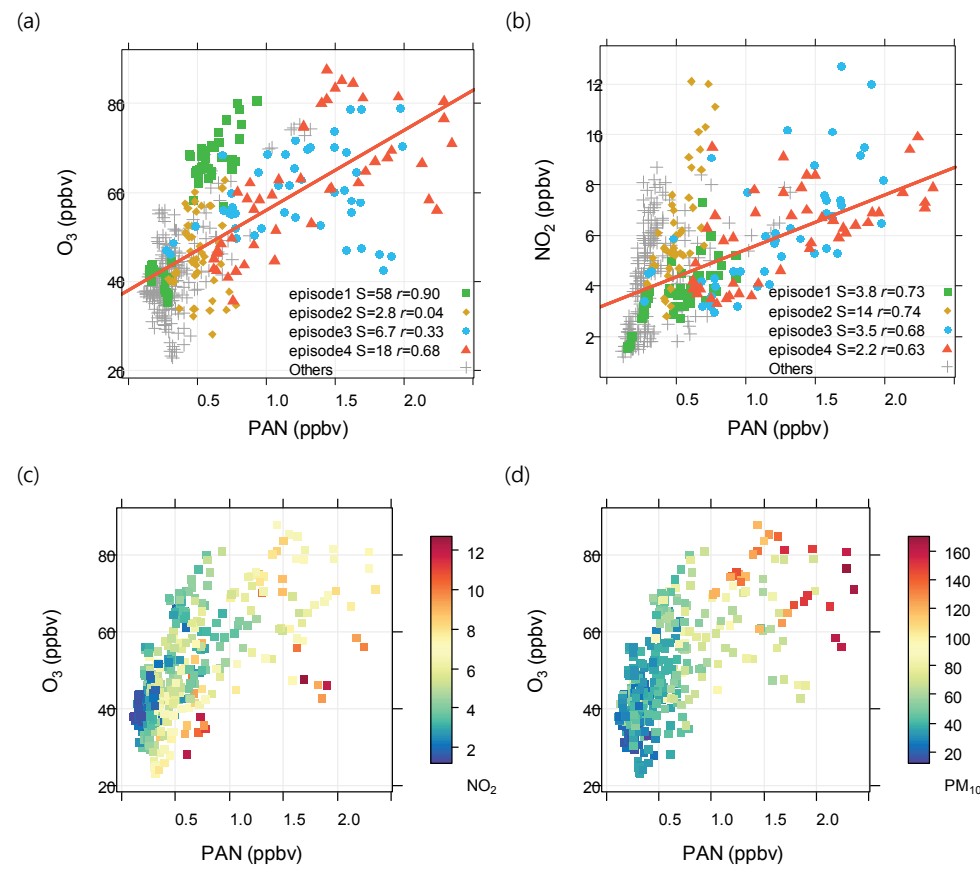


Figure 5.








Figure 6.



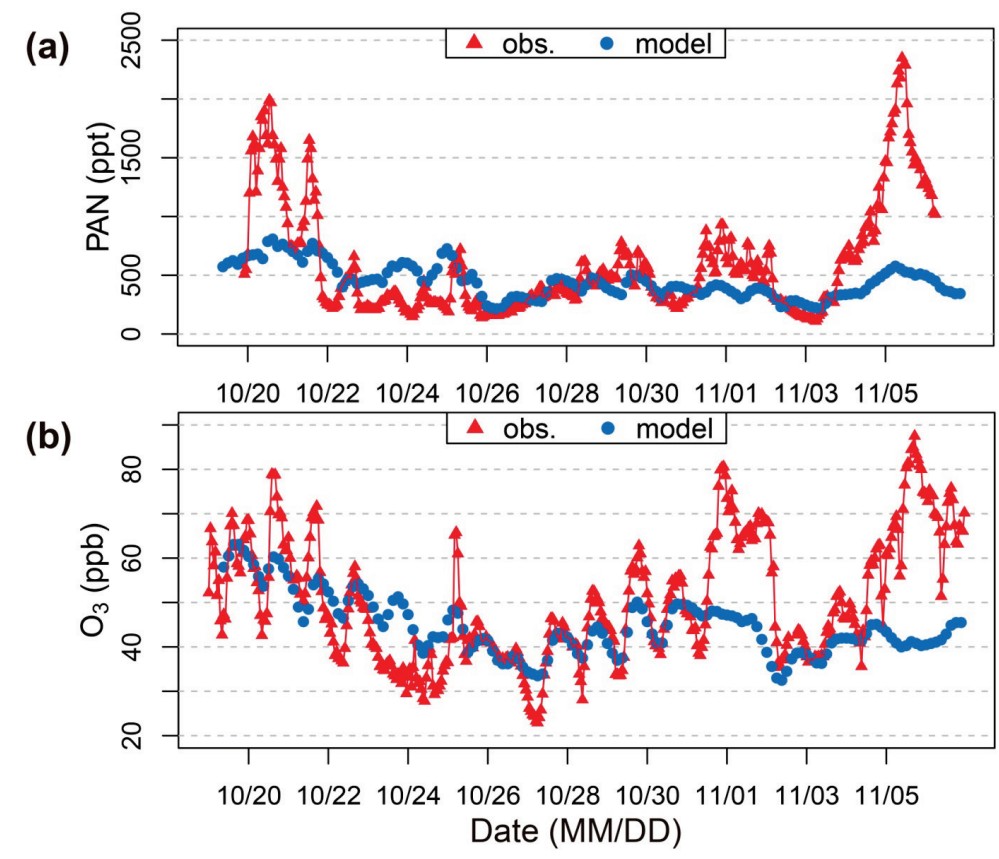


Figure 7.