# Peer review of "Decoupling peroxyacetyl nitrate from ozone in Chinese outflows observed at Gosan Climate Observatory"

_Atmospheric Chemistry and Physics, 2016_

## Referee Comment (RC1) · Anonymous Referee #1 · 22 Feb 2017

A review of "Decoupling peroxyacetyl nitrate from ozone in Chinese outflows observed at Gosan Climate Observatory" by Han et al. submitted to ACP

General comments:

The paper is based on continuous observations of PAN and other air pollutants at Gosan Observatory on Jeju Island in Korea. First, the authors described time series of these species, analyzed diurnal cycles, and then focused on four characteristic episodes, followed by detailed correlative analysis of PAN to other species, in particular O3 and/or PM10 and PM2.5 components. The authors found better correlation of PAN with PM10 than with O3 in pollution plumes transported from China, while the correlation with O3 is also reasonably good. The authors suggested this happened in biomass

burning plumes. Overall, the paper is well organized and written. This is an interesting data and new results, and I enjoyed reading the paper. This paper would be a nice piece of work contributing to the community. However, the paper needs some clarifications and a bit more details in the analysis. I have several comments and suggestions for the authors to put into the revised paper.

Major comments:

The authors highlighted PM10. I wonder why PM10 not PM2.5, as I would expect the same or even better correlation with PM2.5. Is this just because the authors had the PM10 mass concentration data, not PM2.5 mass concentration data? If so, the explicit statement of PM10 might be misleading, as the readers would think that PAN was specifically correlated with larger particles. Otherwise, the authors can simply mention aerosols without size information. The authors need clarifications on this point.

The authors finding of good correlation of PAN to aerosols is interesting and this is obvious from the data, but on the other hand the correlation of PAN to O3 is also good. So, I think decoupling of PAN from O3 is a bit too strong statement, and furthermore, PAN as a potential indicator of overall aerosol formation aged air masses impacted by biomass burning (so I guess smaller particles like PM2.5) also sounds too strong, with only the observed good correlative behaviors. The authors would need to put more analysis or interpretation on the underlying mechanisms to make this statement more robust.

Specific comments:

Abstract: PM10 and PM2.5 OC and EC. . . Does it mean PM10 mass concentration, OC and EC in PM2.5? Please clarify.

In Introduction (Page 3) and Results (Page 6), the authors mentioned previous measurements. However, some important references are missing for both urban/suburban and remote sites in East Asia, and I would note below-referenced work made in Japan.

[Figure]

As far as I read the papers, Tanimoto et al. JGR (1999) reported PAN measurements in Tokyo to be 0.6 ppb in November, and Tanimoto et al. JGR (2002) reported approx. 0.5 ppb in spring at Rishiri Island in northern Japan.

Page 6, Line 138-140: what about correlation to PM2.5?

Page 7, Line 147-149: PAN is not always coupled with O3... Again, the authors used the word "decoupling" and implied PAN and O3 are not correlated at all, but in fact they are reasonably correlated and the PAN-aerosol correlation was just better that PAN-O3. Do you consider to rephrase the statement?

Page 8, Line 167-168: elevated concentrations of O3 and/or PAN at night. This is a bit ambiguous. From Figure 3, I can see O3 is elevated but PAN is not. Please clarify or justify.

In Sections 4.2 and 4.3, the authors discuss fast and slow transport, respectively. I would suggest to put some estimates on time-scales (how fast and how slow these transport episodes were). In Page 8, Line 181-185, the authors mentioned PAN and O3 formation in the outflow during transport. However, Tanimoto et al. SOLA (2008) paper reported negligible O3 enhancement in fast transported plumes from biomass burning in Siberia.

Page 12, Line 266–: The paragraph on CAM-chem can be moved to the Experimental Section.

All figures: I can see the x-axis is local time, but please clarify.

Figure 4: What about adding the same figure for PM10?

Figure 6: These figures are highlight of this paper. However, I see sub-figures d and f are somewhat duplicating and I would delete these.

References:

Tanimoto, H., J. Hirokawa, Y. Kajii, and H. Akimoto, A new measurement technique of

peroxyacetyl nitrate at parts per trillion by volume levels: Gas chromatography/negative ion chemical ionization mass spectrometry, J. Geophys. Res., 104(17), 21,343-21,354, 1999.

Tanimoto, H., H. Furutani, S. Kato, J. Matsumoto, Y. Makide, and H. Akimoto, Seasonal cycles of ozone and oxidized nitrogen species in northeast Asia, 1, Impact of regional climatology and photochemistry observed during RISOTTO 1999-2000, J. Geophys. Res., 107(D24), 4747, doi:10.1029/2001JD001496, 2002.

Tanimoto, H., K. Matsumoto, and M. Uematsu, Ozone–CO correlations in Siberian wildfire plumes observed at Rishiri Island, SOLA, 4, 65-68, doi:10.2151/sola.2008-017, 2008.
* * *

---

## Referee Comment (RC2) · Anonymous Referee #2 · 3 Apr 2017

Summary: This paper presents in situ observations of PAN and supporting gas and aerosols from the Gosan Climate Observatory. The analysis focuses on attributing and differentiating four episodes of elevated PAN. Given relatively sparse PAN data, and continued interest in the role of PAN in the export of pollution from East Asia, publication of this data is of great interest. However, I recommend substantial revisions before the paper is published in ACP. I recommend shortening the paper substantially, while also increasing the methodological details. Rather than focusing on the "decoupling of ozone and PAN", which is not actually novel, particularly in biomass burning plumes and in instances of long range transport. I think it would be better to simply present this as an attribution of four different instances of elevated PAN.

[Figure]

Major Comments: Lines 89 - 114: The experimental details are insufficient and should be greatly expanded to include appropriate references for every measurement used; details on calibration technique and frequency should also be reported. The detection limits and uncertainty should also be included. More detail is particularly important for the NO and NO2 measurements, which are used in the PAN lifetime estimates. The NO2 measurement is likely to include other NOy species. This should be noted and discussed. For PAN, please include additional information on how long PAN was in the instrument, what was the inlet length, etc. 100 pptv is a very high detection limit for an in situ measurement. The calibration technique is also infrequently used. Please explain why 850 m was chosen for the trajectory initialization height.

Line 129: Start a new paragraph. It is unclear what 'haze" means. I believe this is simply a period with elevated aerosol concentrations

Section 4.1: I recommend removing this section and Figure 3. This section and that Figure are confusing, with low science content.

Section 4.2: I recommend reordering this section to discuss clear similarities and differences between these episodes. Lines 192-193: It is odd to claim something is decoupled if it is correlated with something.

Section 4.3: I recommend reordering this section to discuss clear similarities and differences between these episodes. I am skeptical of the PAN lifetime calculations given the unknown quality of the NO2 measurement. More discussion is required. Line 240 – 241: I find these sentences quite confusing, and I do not understand the logic.

Lines 266 – 275: Model methods belong in the methods section. This section was clearly written by a single coauthor and has a different quality of English than the rest of the paper.

Line 277-278: I think the main point of this section is to show that the model severely underestimates PAN, particularly when biomass burning is a source. Thus shorten to

just make this point.

Figure Comments: Figure 1: The ozone line is not visible when Figure 1 is printed. The wind barbs are too small.

Figure 2: It hard to actually get hour-to-hour differences from this figure. Ideas for improvement include making log y-axes. Or a 2x2 figure, rather than 1x4. Figure 3: remove Figure 4: Remove redundant longitude line labels. "O3" and "NO2" overlap the bottom of the scale. All species need units on figure. No vertical information is shown for these trajectories. Did any intercept the ground – were they removed? Figure 5: a/b) why is the fit only shown for Episode 4? Figure 6: It is probably not necessary to show the correlation with all aerosol constituents. Again, why is only the Episode 4 fit shown. Figure 7: Add labels for events as in Figure 1.

Other Minor Comments:

Line 29: Extra E in lat/lon

Line 61: Awkward sentence

Throughout: exchange "concentration" for "mixing ratio" when referring to the abundance of gas phase species

Line 68: A few pptv of PAN is observed in the most remote locations. Change to "In the most remote..."

Line 71: NOx has recently declined in China (e.g. Liu et al., 2016, Environmental Research Letters, Volume 11, Number 11)

Line 71- 81: It would be good to mention in this introduction that PAN is only sparingly soluble. Since there is an eventual discussion of biofuel/ biomass burning, more recent relevant references for elevated PAN plumes attributed to biomass burning downwind of Russia and East Asia include:

Zhu, L., V. H. Payne, T. W. Walker, J. R. Worden, Z. Jiang, S. S. Kulawik, and E. V.

Fischer (2017), PAN in the eastern Pacific free troposphere: A satellite view of the sources, seasonality, interannual variability, and timeline for trend detection, J. Geophys. Res. Atmos., 122, doi:10.1002/2016JD025868.

Zhu, L., E. V. Fischer, V. H. Payne, J. R. Worden, and Z. Jiang (2015), TES observations of the interannual variability of PAN over Northern Eurasia and the relationship to springtime fires, Geophys. Res. Lett.,DOI:10.1002/2015GL065328.

---

## Author Comment (AC1) · 22 May 2017

Correspondence to Referee #1

Thank you very much for your constructive comments. The response for each comment is given below and manuscript was revised accordingly.

Major comments The authors highlighted PM10. I wonder why PM10 not PM2.5, as I would expect the same or even better correlation with PM2.5. Is this just because the authors had the PM10 mass concentration data, not PM2.5 mass concentration data? If so, the explicit statement of PM10 might be misleading, as the readers would think that PAN was specifically correlated with larger particles. Otherwise, the authors can

simply mention aerosols without size information. The authors need clarifications on this point. The authors finding of good correlation of PAN to aerosols is interesting and this is obvious from the data, but on the other hand the correlation of PAN to O3 is also good. So, I think decoupling of PAN from O3 is a bit too strong statement, and furthermore, PAN as a potential indicator of overall aerosol formation aged air masses impacted by biomass burning (so I guess smaller particles like PM2.5) also sounds too strong, with only the observed good correlative behaviors. The authors would need to put more analysis or interpretation on the underlying mechanisms to make this statement more robust.

1) PM2.5

During the experiment, PM2.5 was measured under guidance of NIER (National Institute of Environmental Research). However, we were not allowed to report PM2.5 measurement results publically because PM2.5 measurement officially began in 2015. That is why we used PM10 instead of PM2.5 in the manuscript, even though PM2.5 measurements were included for data analysis. These days, however, the levels of PM2.5 has become one of the top issues of Korea and accordingly, it gets much easier to get an access to air quality data. Thus, PM10 was replaced with PM2.5 and relevant statements were changed in revised manuscript. .

2) Decoupling

In urban air, PAN is proportionally increased with O3, leading to good correlation between their daily maxima (e.g., Lee et al., 2008; Zhang et al. 2009) as well as their single measurements.

[Zhang et al., 2009] [Lee et al., 2008]

Zhang, J. M., Wang, T., Ding, A. J., Zhou, X. H., Xue, L. K., Poon, C. N., Wu, W. S., Gao, J., Zuo, H. C., Chen, J. M., Zhang, X. C., and Fan, S. J.: Continuous measurement of peroxyacetyl nitrate (PAN) in suburban and remote areas of western China, Atmos.

[Figure]

Environ., 43, 228-237, 2009.

Lee, G., Jang, Y., Lee, H., Han, J.-S., Kim, K.-R., and Lee, M.: Characteristic behavior of peroxyacetyl nitrate (PAN) in Seoul megacity, Korea, Chemosphere, 73, 619-628, 10.1016/j.chemosphere.2008.05.060, 2008.

In this study, there is reasonable correlation between hourly O3 and PAN measurements as shown in Figure 5a. However, daily PAN and O3 maxima were not proportionally increased particularly in noticeable episodes associated with Chinese outflows, which is presented in Figure 3. Thus, this episode-specific phenomenon is referred to as 'decoupling' in the manuscript. It is also specified in the title. Nonetheless, we found some confusing statement and a special care was given to eliminate ambiguity. For example, the sentence in the line 192-194 (submitted manuscript) was rewritten as follows.

Therefore, the nighttime maximum of O3 can be attributed to the export of O3 from megacities in China. It causes PAN to be decoupled from O3 because PAN levels remained low, even though there was good correlation between the two species.

3) PAN as fine aerosol (PM2.5)

In revised manuscript, the last statement of the abstract was reworded as follows:

"This study highlights PAN decoupling with O3 in Chinese outflows and suggests PAN as a useful indicator for diagnosing continental outflows and assessing their perturbation on regional air quality in northeast Asia." "

Specific comments

1. Abstract: PM10 and PM2.5 OC and EC... Does it mean PM10 mass concentration, OC and EC in PM2.5? Please clarify.

As mentioned above, PM10 was replaced with PM2.5 and the relevant figures and statements were all changed in the revised manuscript.

2. In Introduction (Page 3) and Results (Page 6), the authors mentioned previous measurements. However, some important references are missing for both urban/suburban and remote sites in East Asia, and I would note below-referenced work made in Japan. As far as I read the papers, Tanimoto et al. JGR (1999) reported PAN measurements in Tokyo to be 0.6 ppb in November, and Tanimoto et al. JGR (2002) reported approx. 0.5 ppb in spring at Rishiri Island in northern Japan.

They are all added to the revised manuscript in introduction (page 3, line 63-65) and results (page 6, line 118-126).

3. Page 6, Line 138-140: what about correlation to PM2.5?

As PM10 was replaced with PM2.5, relevant statements, Table 1, and Figures (1, 2, 4, 5, 6) were all changed in the revised manuscript. The following figure shows the relationship between PM10 and PM2.5 mass concentrations (slope and r2 values inside) that was not included in the manuscript.

4. Page 7, Line 147-149: PAN is not always coupled with O3. . . Again, the authors used the word "decoupling" and implied PAN and O3 are not correlated at all, but in fact they are reasonably correlated and the PAN-aerosol correlation was just better that PAN-O3. Do you consider to rephrase the statement?

Please see the response for major comments.

5. Page 8, Line 167-168: elevated concentrations of O3 and/or PAN at night. This is a bit ambiguous. From Figure 3, I can see O3 is elevated but PAN is not. Please clarify or justify.

The highest PAN concentrations were observed during the day in haze days, which is shown in Figure 3. While PAN reached the maximum during the day on Oct 20 and Nov 5, their concentrations were increased from the previous day through the night. This sentence was added to the text to make it clear.

6. In Sections 4.2 and 4.3, the authors discuss fast and slow transport, respectively.

I would suggest to put some estimates on time-scales (how fast and how slow these transport episodes were). In Page 8, Line 181-185, the authors mentioned PAN and O3 formation in the outflow during transport. However, Tanimoto et al. SOLA (2008) paper reported negligible O3 enhancement in fast transported plumes from biomass burning in Siberia.

The time scale is given to the text. It would take 2 days at most for air being transported from Beijing area to Gosan. In contrast, it took about 3 days during haze episode due to the stagnant condition. Although Beijing area is farther from Jeju than Jiangsu province, the Beijing plume reached faster than the biomass combustion impacted air coming from Jinangsu Province.

Tanimoto et al. (2008) compared Boreal biomass burning plumes observed in Rishiri Island and reported that O3 production was not substantial in fresh air masses. In this study, the Beijing plume captured in Gosan is thought to be relatively fresh (less than 2 days) and could be no considerable ozone production during transport. In addition, there was the best correlation between O3 and PAN. Thus, the following sentence was removed in the revised manuscript.

"In this episode, PAN and O3 could have been formed in urban areas or produced in the outflow while being transported."

Page 12, Line 266–: The paragraph on CAM-chem can be moved to the Experimental Section.

The paragraph was moved to the experimental section.

7. All figures: I can see the x-axis is local time, but please clarify.

It is clarified in figure captions (Figure 1& 7).

8. Figure 4: What about adding the same figure for PM10?

As suggested, the color coded trajectories by PM2.5 concentrations is added to Figure

4.

9. Figure 6: These figures are highlight of this paper. However, I see sub-figures d and f are somewhat duplicating and I would delete these.

Figure 6f was removed. Figure 6d shows the enhancement of OC against EC (âŰşOC/âŰşEC) for episode 4, indicating the influence of biomass combustion. Therefore, 6c was removed instead of 6d.

[Figure]

Correspondence to Referee #1

Thank you very much for your constructive comments. The response for each comment is given below and manuscript was revised accordingly.

Major comments
The authors highlighted PM10. I wonder why PM10 not PM2.5, as I would expect the same or even better correlation with PM2.5. Is this just because the authors had the PM10 mass concentration data, not PM2.5 mass concentration data? If so, the explicit statement of PM10 might be misleading, as the readers would think that PAN was specifically correlated with larger particles. Otherwise, the authors can simply mention aerosols without size information. The authors need clarifications on this point.
The authors finding of good correlation of PAN to aerosols is interesting and this is obvious from the data, but on the other hand the correlation of PAN to O3 is also good. So, I think decoupling of PAN from O3 is a bit too strong statement, and furthermore, PAN as a potential indicator of overall aerosol formation aged air masses impacted by biomass burning (so I guess smaller particles like PM2.5) also sounds too strong, with only the observed good correlative behaviors. The authors would need to put more analysis or interpretation on the underlying mechanisms to make this statement more robust.

1) PM2.5

During the experiment, PM2.5 was measured under guidance of NIER (National Institute of Environmental Research). However, we were not allowed to report PM2.5 measurement results publically because PM2.5 measurement officially began in 2015. That is why we used PM10 instead of PM2.5 in the manuscript, even though PM2.5 measurements were included for data analysis. These days, however, the levels of PM2.5 has become one of the top issues of Korea and accordingly, it gets much easier to get an access to air quality data. Thus, PM10 was replaced with PM2.5 and relevant statements were changed in revised manuscript.  .

2) Decoupling

In urban air, PAN is proportionally increased with O$_3$, leading to good correlation between their daily maxima (e.g., Lee et al., 2008; Zhang et al. 2009) as well as their single measurements.

**Fig. 1.**

---

## Author Comment (AC2) · 22 May 2017

Correspondence to Referee #2

Thank you very much for your constructive comments. The response for each comment is given below and manuscript was revised accordingly.

Major comments Lines 89 - 114: The experimental details are insufficient and should be greatly expanded to include appropriate references for every measurement used; details on calibration technique and frequency should also be reported. The detection limits and uncertainty should also be included. More detail is particularly important for the NO and NO2 measurements, which are used in the PAN lifetime estimates. The

[Figure]

NO2 measurement is likely to include other NOy species. This should be noted and discussed. For PAN, please include additional information on how long PAN was in the instrument, what was the inlet length, etc. 100 pptv is a very high detection limit for an in situ measurement. The calibration technique is also infrequently used. Please explain why 850 m was chosen for the trajectory initialization height.

1) Experiment section was revised with more detailed information on measurement methods of PAN and NOx. Please see Section 2.Experiement in the revised manuscript.

2) It is well known that molybdenum convertor gives positive artifact to NO2 measurement due to the interference of NOy species including HNO3, PAN, and HONO. Unfortunately, NOy measurements are not available at Gosan. However, the concentration of NOx is generally low at Gosan climate observatory as a regional background site (e.g., Lee et al, 2007, Lee et al., 2014). Therefore, the influence of NOy would be negligible.

M. Lee, et al., Origins and chemical characteristics of fine aerosols during the northeastern Asia regional experiment (Atmospheric Brown Cloud–East Asia Regional Experiment 2005), JGR, 112, doi:10.1029/2006JD008210, 2007.

H.-J. Lee, et al., Transport of NOx in East Asia identified by satellite and in situ measurements and Lagrangian particle dispersion model simulations, JGR, 119, doi:10.1002/2013JD021185, 2014.

3) Trajectories are given for 850 m altitude in the manuscript. However, we checked other altitudes including 500 m and 1000 m and found no tangible difference among the three altitudes. To represent the surface condition, altitudes between 500 m and 1500 m have been usually selected in this region depending on season.

Specific comments Line 129: Start a new paragraph. It is unclear what 'haze" means. I believe this is simply a period with elevated aerosol concentrations.

[Figure]

As one of the meteorological phenomena, Haze is reported by Korean Meteorological Administration (KMA). The definition is given below.

- Fog: visibility < 1 km & relative humidity > 75 % - Mist: visibility 1∼10 km & relative humidity > 75 % - Haze: visibility 1∼10 km & relative humidity < 75 %

For clarity, the following sentence was added to the text.

"Haze is reported by Korea Meteorological Administration (KMA) as a meteorological phenomenon when visibility is 1∼ 10 km and relative humidity is less than 75 %."

Section 4.1: I recommend removing this section and Figure 3. This section and that Figure are confusing, with low science content.

In urban air, O3 and PAN are generally well correlated with a clear peak in the afternoon. The two species share precursors and produced from their reactions in the atmosphere. Therefore, their maximum values are sensitive to temperature and found to be well correlated (e.g., Lee et al., 2008; Zhang et al. 2009).

[Zhang et al., 2009] [Lee et al., 2008]

Zhang, J. M., Wang, T., Ding, A. J., Zhou, X. H., Xue, L. K., Poon, C. N., Wu, W. S., Gao, J., Zuo, H. C., Chen, J. M., Zhang, X. C., and Fan, S. J.: Continuous measurement of peroxyacetyl nitrate (PAN) in suburban and remote areas of western China, Atmos. Environ., 43, 228-237, 2009.

Lee, G., Jang, Y., Lee, H., Han, J.-S., Kim, K.-R., and Lee, M.: Characteristic behavior of peroxyacetyl nitrate (PAN) in Seoul megacity, Korea, Chemosphere, 73, 619-628, 10.1016/j.chemosphere.2008.05.060, 2008.

Gosan is not an urban site and thus, the tight correlation between PAN and O3 was not expected. Nonetheless, the nighttime maximum of O3 and PAN observed in Chinese outflows was striking and we believe is worth being shown. As you pointed out, it gives confusion because the peaks of O3 and PAN were separated into 4 groups according to

the time of their occurrence. However, the following discussion focused on the highest concentrations and thus, attention needs to be paid only for the two cases of nighttime max. and daytime max, which was clarified in the text by rephrasing the second last sentence in Section 4.1, where "This point" was changed to "These two cases".

Section 4.2: I recommend reordering this section to discuss clear similarities and differences between these episodes. Lines 192-193: It is odd to claim something is decoupled if it is correlated with something.

The following statement was added to the last of Section 4.2.

"These two urban plumes are well contrasted in terms of O3 and NOx levels (Table 1), depending on the degree of aging."

The part of line 192-193 was rewritten as follows:

"Therefore, the nighttime maximum of O3 can be attributed to the export of O3 from megacities in China. It causes PAN to be decoupled from O3 because PAN levels remained low, even though there was good correlation between the two species."

Section 4.3: I recommend reordering this section to discuss clear similarities and differences between these episodes. I am skeptical of the PAN lifetime calculations given the unknown quality of the NO2 measurement. More discussion is required. Line 240 – 241: I find these sentences quite confusing, and I do not understand the logic.

As you suggest, this section was rearranged. Please see Section 4.3. in the revised manuscript

As mention above, more detailed information on measurement method was given to the revised manuscript and the interference would not be substantial.

The statement in Line 240-241 is not necessary and thus, removed in the revised manuscript.

Lines 266 – 275: Model methods belong in the methods section. This section was

clearly written by a single coauthor and has a different quality of English than the rest of the paper.

As you recommended, the model description was moved to the experiment section.

Line 277-278: I think the main point of this section is to show that the model severely underestimates PAN, particularly when biomass burning is a source. Thus shorten to just make this point.

The sentence was rewritten as follows:

"The elevated PAN concentrations during the haze events were underestimated in the model (Oct 20–21 and Nov 4–5), especially when air was impacted by biomass combustion."

Figure Comments: Figure 1: The ozone line is not visible when Figure 1 is printed. The wind barbs are too small.

Figure 1 was modified according to your suggestions.

Figure 2: It hard to actually get hour-to-hour differences from this figure. Ideas for improvement include making log y-axes. Or a 2x2 figure, rather than 1x4.

Figure 2 was modified according to your suggestions.

Figure 3: remove

Pease see the response for section 4.1 in page 2.

Figure 4: Remove redundant longitude line labels. "O3" and "NO2" overlap the bottom of the scale. All species need units on figure. No vertical information is shown for these trajectories. Did any intercept the ground – were they removed?

Figure 4 was modified according to your suggestions. The altitude information was added, too. Trajectories that intercept the ground were removed.

Figure 5: a/b) why is the fit only shown for Episode 4? Figure 6: It is probably not

necessary to show the correlation with all aerosol constituents. Again, why is only the Episode 4 fit shown.

In Figure 5 and 6, regression lines were added to the other episodes.

Figure 7: Add labels for events as in Figure 1.

Episode lables were added to Figure 7.

Other Minor Comments: Line 29: Extra E in lat/lon

Typo was corrected.

Line 61: Awkward sentence

The error was corrected as follows:

"Thus, PAN can be an indicator of NOy concentration in the free troposphere and a guide for the long-range transport of NOx in remote regions"

Throughout: exchange "concentration" for "mixing ratio" when referring to the abundance of gas phase species

It was corrected.

Line 68: A few pptv of PAN is observed in the most remote locations. Change to "In the most remote. . ."

It was corrected.

Line 71: NOx has recently declined in China (e.g. Liu et al., 2016, Environmental Research Letters, Volume 11, Number 11)

The relevant part was rewritten as follows:

"Although NOx concentration has recently declined in China (Gu et al., 2013; Liu et al., 2016a; Krotkov et al., 2016), NOx and VOCs have gradually increased in East Asia, particularly China during the last couple of decades (Akimoto, 2003; Liu et al., 2010;

Ohara et al., 2007; Zhao et al., 2013). It led to an increase in the concentrations of photochemical byproducts such as PAN and O3 not only in East Asia (Liu et al., 2010; Wang et al., 2010; Zhang et al., 2009; Zhang et al., 2011; Zhang et al., 2014) but also in North America (Fischer et al., 2010; Fischer et al., 2011; Jaffe et al., 2007; Zhang et al., 2008)."

Line 71- 81: It would be good to mention in this introduction that PAN is only sparingly soluble. Since there is an eventual discussion of biofuel/ biomass burning, more recent relevant references for elevated PAN plumes attributed to biomass burning downwind of Russia and East Asia include.

The statement regarding PAN solubility was added to the text as follows:

"Besides, PAN is less soluble compared to nitric acid and is more easily transported to the free troposphere after it is released from scavenge in lower temperature (e.g., Zhu et al., 2017)."

References were added and the following statement was added to the text.

"Recent satellite studies have also observed the increased PAN in plumes associated with anthropogenic emissions in eastern China and boreal fires in Siberia (Zhu et al., 2015; Zhu et al., 2017)."
* * *
Correspondence to Referee #2

Thank you very much for your constructive comments. The response for each comment is given below and manuscript was revised accordingly.

Major comments
Lines 89 - 114: The experimental details are insufficient and should be greatly expanded to include appropriate references for every measurement used; details on calibration technique and frequency should also be reported. The detection limits and uncertainty should also be included. More detail is particularly important for the NO and NO2 measurements, which are used in the PAN lifetime estimates. The NO2 measurement is likely to include other NOy species. This should be noted and discussed. For PAN, please include additional information on how long PAN was in the instrument, what was the inlet length, etc. 100 pptv is a very high detection limit for an in situ measurement. The calibration technique is also infrequently used. Please explain why 850 m was chosen for the trajectory initialization height.

1) Experiment section was revised with more detailed information on measurement methods of PAN and NOx. Please see Section 2.Experiement in the revised manuscript.

2) It is well known that molybdenum convertor gives positive artifact to $NO_2$ measurement due to the interference of $NO_y$ species including $HNO_3$, PAN, and HONO. Unfortunately, $NO_y$ measurements are not available at Gosan. However, the concentration of $NO_x$ is generally low at Gosan climate observatory as a regional background site (e.g., Lee et al, 2007, Lee et al., 2014). Therefore, the influence of $NO_y$ would be negligible.

M. Lee, et al., Origins and chemical characteristics of fine aerosols during the northeastern Asia regional experiment (Atmospheric Brown Cloud–East Asia Regional Experiment 2005), JGR, 112, doi:10.1029/2006JD008210, 2007.

H.-J. Lee, et al., Transport of NOx in East Asia identified by satellite and in situ measurements and Lagrangian particle dispersion model simulations, JGR, 119, doi:10.1002/2013JD021185, 2014.

3) Trajectories are given for 850 m altitude in the manuscript. However, we checked other altitudes including 500 m and 1000 m and found no tangible difference among the three altitudes. To represent the surface condition, altitudes between 500 m and 1500 m have been usually selected in this region depending on season.

Specific comments
Line 129: Start a new paragraph. It is unclear what 'haze" means. I believe this is simply a period with elevated aerosol concentrations.

**Fig. 1.**

[Figure]

---

## Author Comment (AC3) · 22 May 2017

Correspondence to Referee #1

Thank you very much for your constructive comments. The response for each comment is given below and manuscript was revised accordingly.

Major comments The authors highlighted PM10. I wonder why PM10 not PM2.5, as I would expect the same or even better correlation with PM2.5. Is this just because the authors had the PM10 mass concentration data, not PM2.5 mass concentration data? If so, the explicit statement of PM10 might be misleading, as the readers would think that PAN was specifically correlated with larger particles. Otherwise, the authors can

[Figure]

simply mention aerosols without size information. The authors need clarifications on this point. The authors finding of good correlation of PAN to aerosols is interesting and this is obvious from the data, but on the other hand the correlation of PAN to O3 is also good. So, I think decoupling of PAN from O3 is a bit too strong statement, and furthermore, PAN as a potential indicator of overall aerosol formation aged air masses impacted by biomass burning (so I guess smaller particles like PM2.5) also sounds too strong, with only the observed good correlative behaviors. The authors would need to put more analysis or interpretation on the underlying mechanisms to make this statement more robust.

1) PM2.5

During the experiment, PM2.5 was measured under guidance of NIER (National Institute of Environmental Research). However, we were not allowed to report PM2.5 measurement results publically because PM2.5 measurement officially began in 2015. That is why we used PM10 instead of PM2.5 in the manuscript, even though PM2.5 measurements were included for data analysis. These days, however, the levels of PM2.5 has become one of the top issues of Korea and accordingly, it gets much easier to get an access to air quality data. Thus, PM10 was replaced with PM2.5 and relevant statements were changed in revised manuscript. .

2) Decoupling

In urban air, PAN is proportionally increased with O3, leading to good correlation between their daily maxima (e.g., Lee et al., 2008; Zhang et al. 2009) as well as their single measurements.

[Zhang et al., 2009] [Lee et al., 2008]

Zhang, J. M., Wang, T., Ding, A. J., Zhou, X. H., Xue, L. K., Poon, C. N., Wu, W. S., Gao, J., Zuo, H. C., Chen, J. M., Zhang, X. C., and Fan, S. J.: Continuous measurement of peroxyacetyl nitrate (PAN) in suburban and remote areas of western China, Atmos.

Environ., 43, 228-237, 2009.

Lee, G., Jang, Y., Lee, H., Han, J.-S., Kim, K.-R., and Lee, M.: Characteristic behavior of peroxyacetyl nitrate (PAN) in Seoul megacity, Korea, Chemosphere, 73, 619-628, 10.1016/j.chemosphere.2008.05.060, 2008.

In this study, there is reasonable correlation between hourly O3 and PAN measurements as shown in Figure 5a. However, daily PAN and O3 maxima were not proportionally increased particularly in noticeable episodes associated with Chinese outflows, which is presented in Figure 3. Thus, this episode-specific phenomenon is referred to as 'decoupling' in the manuscript. It is also specified in the title. Nonetheless, we found some confusing statement and a special care was given to eliminate ambiguity. For example, the sentence in the line 192-194 (submitted manuscript) was rewritten as follows.

Therefore, the nighttime maximum of O3 can be attributed to the export of O3 from megacities in China. It causes PAN to be decoupled from O3 because PAN levels remained low, even though there was good correlation between the two species.

3) PAN as fine aerosol (PM2.5)

In revised manuscript, the last statement of the abstract was reworded as follows:

"This study highlights PAN decoupling with O3 in Chinese outflows and suggests PAN as a useful indicator for diagnosing continental outflows and assessing their perturbation on regional air quality in northeast Asia." "

Specific comments

1. Abstract: PM10 and PM2.5 OC and EC... Does it mean PM10 mass concentration, OC and EC in PM2.5? Please clarify.

As mentioned above, PM10 was replaced with PM2.5 and the relevant figures and statements were all changed in the revised manuscript.

2. In Introduction (Page 3) and Results (Page 6), the authors mentioned previous measurements. However, some important references are missing for both urban/suburban and remote sites in East Asia, and I would note below-referenced work made in Japan. As far as I read the papers, Tanimoto et al. JGR (1999) reported PAN measurements in Tokyo to be 0.6 ppb in November, and Tanimoto et al. JGR (2002) reported approx. 0.5 ppb in spring at Rishiri Island in northern Japan.

They are all added to the revised manuscript in introduction (page 3, line 63-65) and results (page 6, line 118-126).

3. Page 6, Line 138-140: what about correlation to PM2.5?

As PM10 was replaced with PM2.5, relevant statements, Table 1, and Figures (1, 2, 4, 5, 6) were all changed in the revised manuscript. The following figure shows the relationship between PM10 and PM2.5 mass concentrations (slope and r2 values inside) that was not included in the manuscript.

4. Page 7, Line 147-149: PAN is not always coupled with O3... Again, the authors used the word "decoupling" and implied PAN and O3 are not correlated at all, but in fact they are reasonably correlated and the PAN-aerosol correlation was just better that PAN-O3. Do you consider to rephrase the statement?

Please see the response for major comments.

5. Page 8, Line 167-168: elevated concentrations of O3 and/or PAN at night. This is a bit ambiguous. From Figure 3, I can see O3 is elevated but PAN is not. Please clarify or justify.

The highest PAN concentrations were observed during the day in haze days, which is shown in Figure 3. While PAN reached the maximum during the day on Oct 20 and Nov 5, their concentrations were increased from the previous day through the night. This sentence was added to the text to make it clear.

6. In Sections 4.2 and 4.3, the authors discuss fast and slow transport, respectively.

I would suggest to put some estimates on time-scales (how fast and how slow these transport episodes were). In Page 8, Line 181-185, the authors mentioned PAN and O3 formation in the outflow during transport. However, Tanimoto et al. SOLA (2008) paper reported negligible O3 enhancement in fast transported plumes from biomass burning in Siberia.

The time scale is given to the text. It would take 2 days at most for air being transported from Beijing area to Gosan. In contrast, it took about 3 days during haze episode due to the stagnant condition. Although Beijing area is farther from Jeju than Jiangsu province, the Beijing plume reached faster than the biomass combustion impacted air coming from Jinangsu Province.

Tanimoto et al. (2008) compared Boreal biomass burning plumes observed in Rishiri Island and reported that O3 production was not substantial in fresh air masses. In this study, the Beijing plume captured in Gosan is thought to be relatively fresh (less than 2 days) and could be no considerable ozone production during transport. In addition, there was the best correlation between O3 and PAN. Thus, the following sentence was removed in the revised manuscript.

"In this episode, PAN and O3 could have been formed in urban areas or produced in the outflow while being transported."

Page 12, Line 266–: The paragraph on CAM-chem can be moved to the Experimental Section.

The paragraph was moved to the experimental section.

7. All figures: I can see the x-axis is local time, but please clarify.

It is clarified in figure captions (Figure 1& 7).

8. Figure 4: What about adding the same figure for PM10?

As suggested, the color coded trajectories by PM2.5 concentrations is added to Figure

4.

9. Figure 6: These figures are highlight of this paper. However, I see sub-figures d and f are somewhat duplicating and I would delete these.

Figure 6f was removed. Figure 6d shows the enhancement of OC against EC (âŰşOC/âŰşEC) for episode 4, indicating the influence of biomass combustion. Therefore, 6c was removed instead of 6d.

Please also note the supplement to this comment:
http://www.atmos-chem-phys-discuss.net/acp-2016-1107/acp-2016-1107-AC3-supplement.pdf

**Supplement:**

**Decoupling peroxyacetyl nitrate from ozone in Chinese outflows observed at Gosan Climate Observatory**

**Jihyun Han[1*], Meehye Lee[1], Gangwoong Lee[2], Louisa K. Emmons[3]**

[1]Department of Earth and Environmental Sciences, Korea University, Seoul, Republic of Korea

[2]Department of Environmental Science, Hankuk University of Foreign Studies, Yongin, Republic of Korea

[3]Atmospheric Chemistry Observations and Modeling Laboratory, National Center for Atmospheric Research (NCAR), Boulder, CO, USA

*now at: Korea Environment Institute, Sejong, Republic of Korea

Correspondence to: M. Lee (meehye@korea.ac.kr )

Submitted to Atmospheric Chemistry and Physics

December 2016

**Abstract**

We measured peroxyacetyl nitrate (PAN) and other reactive species such as $O_3$, $NO_2$, CO, and $SO_2$ with aerosols including mass, organic carbon (OC), and elemental carbon (EC) in $PM_{2.5}$ and $K^+$ in $PM_{1.0}$ at Gosan Climate Observatory in Korea (33.17°N, 126.10°E) during October 10 to November 6, 2010. PAN was determined through fast gas chromatography with luminol chemiluminescence detection at 425 nm every 2 min. The PAN concentrations ranged from 0.1 (detection limit) to 2.4 ppbv with a mean of 0.6 ppbv. For all measurements, PAN was unusually better correlated with $PM_{2.5}$ (Pearson correlation coefficient, $\gamma = 0.79$) than with $O_3$ ($\gamma = 0.67$). In particular, the $O_3$ level was highly elevated with $SO_2$ at midnight, along with a typical midday peak when air was transported rapidly from the Beijing areas. The PAN enhancement was most noticeable during the occurrence of haze under stagnant conditions. In Chinese outflows slowly transported over the Yellow Sea, PAN gradually increased up to 2.4 ppbv at night, in excellent correlation with a concentration increase of $PM_{2.5}$ OC and EC, $PM_{2.5}$ mass, and $PM_{1.0}$ $K^+$. The high $K^+$ concentration and OC/EC ratio indicated that the air mass was impacted by biomass combustion. This study highlights PAN decoupling with $O_3$ in Chinese outflows and suggests PAN as a useful indicator for diagnosing continental outflows and assessing their perturbation on regional air quality in northeast Asia.

Key words: PAN, $O_3$, $PM_{2.5}$, Chinese outflow, Haze, Biomass combustion

**1. Introduction**

At the surface, ozone is primarily photochemically produced, and the contribution from the stratosphere is generally small. Ozone is formed through reactions of various precursors such as CO, $CH_4$, volatile organic compounds (VOCs), and $NO_x$ (e.g., Brasseur et al., 1999; Jacob, 2000; Nielsen et al., 1981). Likewise, peroxyacetyl nitrate (PAN) is a secondary product of urban air pollution and a significant oxidant in the atmosphere (e.g., Hansel and Wisthaler, 2000; La Franchi et al., 2009; Lee et al., 2012; Liu et al., 2010; Roberts et al., 2007). PAN is solely produced by the photochemical reaction between the peroxyacetyl radical and nitrogen dioxide, and the peroxyacetyl radical is derived from the OH oxidation or photolysis of VOCs such as acetaldehyde, methylglyoxal, and acetone (e.g., Fischer et al., 2014; La Franchi et al., 2009; Lee et al., 2012). For this reason, PAN is a very useful indicator of photochemical air pollution. As thermal decomposition is a major PAN sink in the troposphere (Beine et al., 1997; Jacob, 2000; Kenley and Hendry, 1982; Talukdar et al., 1995), the lifetime of PAN depends on temperature. For example, the PAN lifetime is ~5 years at −26°C and 1 h at 20°C (Fischer et al., 2010; Zhang et al., 2011). At high altitudes above ~7 km, photolysis becomes the most important loss process for PAN (Talukdar, et al., 1995). Besides, PAN is less soluble compared to nitric acid and is more easily transported to the free troposphere after it is released from scavenge in lower temperature (
[revised manuscript text omitted]

It causes PAN to be decoupled from $O_3$ because PAN levels remained low, even though there was good correlation between the two species. Another night maximum of $O_3$ was recorded on October 29. Note that $NO_x$ was highly elevated during October 28–29 (episode 2) (Fig. 1).

However, $O_3$ level was relatively low, leading to the lowest daily $\Delta O_3/\Delta PAN$ among all episodes. In this case, air masses passed through the Korean Peninsula, carrying low $O_3$ being titrated by high $NO_x$ (Brasseur et al., 1999; Jacobson, 2005). These two urban plumes are well contrasted in terms of $O_3$ and $NO_x$ levels (Table 1), depending on the degree of aging.

[revised manuscript text omitted]
_{2.5}$ mass and PAN, (b) $PM_{2.5}$ OC and PAN, (d) $PM_{2.5}$ OC and EC, and (d) $PM_{1.0}$ $K^+$ and PAN. The lines represent the linear regression for each episode.

Figure 7. Comparison between the observed and calculated (a) PAN and (b) $O_3$ concentrations by CAM-chem model. Time is given in local time and four episodes are shaded.

[Figure]

Figure 1.

[Figure]

Figure 2.

[Figure]

Figure 3.

[Figure]

Figure 4.

[Figure]

Figure 5.

[Figure]

Figure 6.

[Figure]

Figure 7.

---

## Author Response (AR2)

You have not corrected the issues raised by Reviewer #2, associated with the fact that your NO2 measurements are actually NOy-NO measurements. It is clear that the molybdenum oxide conversion technique responds to all NOy species, indeed you use it for your PAN calibration. Yet, you have persisted in referring to the measurement as 'NO2' in your paper, including in the section where you use 'NO2' to estimate PAN thermal decomposition rate. You must correct these thing before your paper can be accepted.

**Correspondence**

It is well known that NO2 measurement using molybdenum converter has interference by NOz compounds. In this study, this artifact would also increase the effective lifetime of PAN. Therefore, the statement regarding the positive artifact of the NO2 measurement was added to Section 2 and this artifact was considered in the estimation of effective lifetime of PAN in Section 4.3. However, we left the term 'NO2' in the manuscript because these concentrations were officially reported as NO2 for Gosan site, which belongs to the national air pollution monitoring network. In the revised manuscript, the modified parts are marked in red and given below.

**Manuscript revision**

1) Page 6 Line 132:

It should be noted that $NO_2$ concentration reported in the present study is actually the sum of $NO_2$ and NOz species due to well-known positive artifact of molybdenum convertor. PAN is one of the major NOz species and the ratio of PAN to $NO_2$ was $12 \pm 7$ % for the whole measurements.

2) Page 12 Line 286:

During the haze event, NO was close to the detection limit, while $NO_2$ was greatly enhanced. Owing to the high $NO_2$/NO ratio, the effective lifetime of PAN increased by $57 \pm 14$ times; this possibly contributed to the gradual increase in PAN through the night on November 4th. For this estimation, PAN concentration was subtracted from the measured $NO_2$ concentration, considering the positive artifact by molybdenum converter in $NO_2$ measurement.